# XDream: Finding preferred stimuli for visual neurons using generative networks and gradient-free optimization

**Will Xiao**[1,2]*, **Gabriel Kreiman**[2,3]

**1** Department of Molecular and Cellular Biology, Harvard University, Cambridge, Massachusetts, United States of America, **2** Center for Brains, Minds, and Machines, Boston, Massachusetts, United States of America, **3** Department of Ophthalmology, Boston Children's Hospital, Boston, Massachusetts, United States of America

* xiaow@fas.harvard.edu

**Data Availability Statement:** All data underlying the findings described in their manuscript have been made available in the publicly available GitHub repository at https://github.com/willwx/XDream.

**Funding:** W.X. and G.K. are supported by the Center for Brains, Minds and Machines funded by

## Abstract

A longstanding question in sensory neuroscience is what types of stimuli drive neurons to fire. The characterization of effective stimuli has traditionally been based on a combination of intuition, insights from previous studies, and luck. A new method termed XDream (**EX**tending **D**eepDream with **r**eal-time **e**volution for **a**ctivation **m**aximization) combined a generative neural network and a genetic algorithm in a closed loop to create strong stimuli for neurons in the macaque visual cortex. Here we extensively and systematically evaluate the performance of XDream. We use ConvNet units as *in silico* models of neurons, enabling experiments that would be prohibitive with biological neurons. We evaluated how the method compares to brute-force search, and how well the method generalizes to different neurons and processing stages. We also explored design and parameter choices. XDream can efficiently find preferred features for visual units without any prior knowledge about them. XDream extrapolates to different layers, architectures, and developmental regimes, performing better than brute-force search, and often better than exhaustive sampling of >1 million images. Furthermore, XDream is robust to choices of multiple image generators, optimization algorithms, and hyperparameters, suggesting that its performance is locally near-optimal. Lastly, we found no significant advantage to problem-specific parameter tuning. These results establish expectations and provide practical recommendations for using XDream to investigate neural coding in biological preparations. Overall, XDream is an efficient, general, and robust algorithm for uncovering neuronal tuning preferences using a vast and diverse stimulus space. XDream is implemented in Python, released under the MIT License, and works on Linux, Windows, and MacOS.

This is a *PLOS Computational Biology* Software paper.

NSF528STC award CCF-1231216 and also by NIH R01EY026025. The funders had no role in study design, data collection and analysis, decision to publish, or preparation of the manuscript. https://www.nsf.gov https://www.nih.gov.

**Competing interests:** The authors have declared that no competing interests exist.

## Introduction

What stimuli excite a neuron, and how can we find them? Consider vision as a paradigmatic example, the selection of stimuli to probe neural activity has shaped the understanding of how visual neurons represent information. It is practically impossible to exhaustively evaluate neuronal responses to images, due to the combinatorially large number of possible images. Instead, investigators have traditionally selected stimuli guided by natural image statistics, behavioral relevance, theoretical postulates about internal representations, intuitions from previous studies, and serendipitous findings. Stimuli selected in this way underlie our current understandings of how circular center-surround receptive fields [1] give rise to orientation tuning [2], then to encoding of more complex shapes such as curvatures [3, 4], and further to selective responses to complex objects such as faces [5–7].

Despite the progress made in understanding visual cortex by testing limited sets of hand-chosen stimuli, these experiments could be missing the true feature preferences of neurons. In other words, there could be other images that drive visual neurons better than those found so far. Such images could lead us to revisit our current descriptions of feature tuning in visual cortex.

A recently introduced method shows promise to begin bridging the gap. Named XDream (e**X**tending **D**eepDream with **r**eal-time **e**volution for **a**ctivation **m**aximization), this method combines a genetic algorithm and a deep generative neural network [8]—both inspired by previous work [9–12]—to evolve images that trigger high activation in neurons [13]. XDream can generate strong stimuli for neurons in macaque inferior temporal (IT) and primary visual cortex (V1).

The performance and design options of XDream have not been thoroughly evaluated, due to the time-intensiveness of neuronal recordings and the difficulty to fully control experimental variables. To overcome these challenges, here we test the performance of XDream using state-of-the-art *in silico* models of visual neurons in lieu of real neurons, in the same spirit of [14]. Specifically, we use convolutional neural networks (ConvNets) pre-trained on visual recognition tasks as an approximation to the computations performed along ventral visual cortex [15–17]. Using these models as a proxy for real neurons allows us to compare synthetic stimuli with a large set of reference images, to evaluate XDream's performance across processing stages, model architectures, and training regimes, to empirically optimize algorithm design and parameter choices in a systematic fashion, and to disentangle the effects of neuronal response stochasticity.

Although there is a rich literature in computer science on feature visualization [18–21], we focus on the more biologically relevant scenario where there is no information about the architecture and weights of the target model, and where we only have access to a few, potentially stochastic, activation values from the neurons. These conditions reflect those prevailing in neuronal recordings and are fundamentally different from the assumptions made in computer science studies.

Under these realistic constraints, we show that XDream still reliably and efficiently uncovers preferred features of units with a wide range of response properties, generalizing to different processing stages within a network, different network architectures, and different training datasets. Furthermore, XDream performed equally well with a wide range of algorithmic and parameter choices. Based on these results, we suggest parameters to use and results that can be expected when using XDream to investigate neuronal tuning properties. Our findings suggest that XDream is a general and robust method for investigating neuronal preferences in visual cortex.

## Design and implementation

**Overview.**   XDream combines an image generator (e.g., the generator in a generative adversarial network), a target neuron (e.g., a unit in a ConvNet), and a non-gradient-based

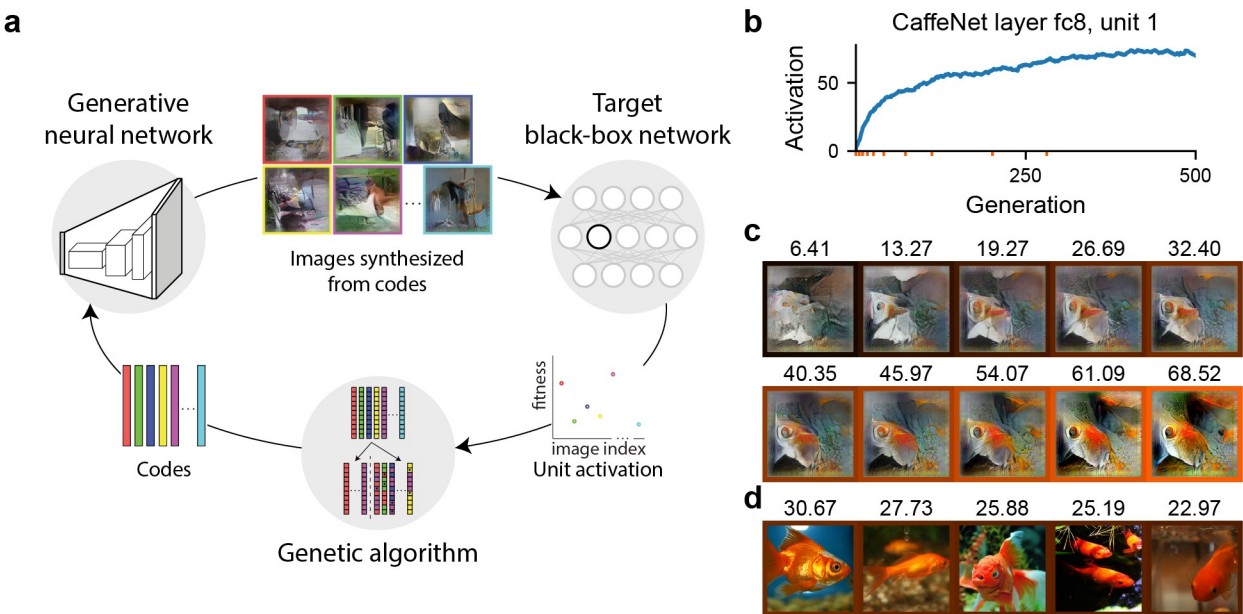

**Fig 1. Overview of the XDream method. a)**, XDream combines an image generator, a target neuron, and a non-gradient-based optimization algorithm. **b,c)**, An example experiment targeting CaffeNet layer fc8, unit 1. **b)**, mean activation achieved over 500 generations, 20 images per generation (10,000 total image presentations). **c)**, Images obtained at a few example generations indicated by minor x-ticks in **b)**. The activation to each image is labeled above the image and indicated by the color of the margin. **d)**, The top 5 images among 10,000 random images from ImageNet (ILSVRC12 dataset, >1.4 M images). The number of random images is matched to the number of images presented during optimization. The top image in all >1.4 M images is shown in Fig 2b.

optimization algorithm (e.g., a genetic algorithm) in a closed loop. In each iteration, the optimization algorithm proposes a set of codes, the image generator synthesizes the codes into images, the images are evaluated by the target neuron to produce one scalar score per image, and the scores are used by the optimization algorithm to propose a new set of codes (Fig 1). Importantly, no optimization gradient is needed from the neuron.

The image generators and optimization algorithms are detailed below. The code is implemented in Python 3 and runs on Linux, Windows, and MacOS, although the former two platforms are required to use GPU acceleration. The main dependency is Caffe [22] (https://caffe.berkeleyvision.org/) or PyTorch (https://pytorch.org/), which are required for neural network computation. Other dependencies are standard Python packages and listed in requirements.txt in the repository, including: numpy, h5py, opencv-python, scipy, and scikit-image.

## Image generators

An image generator is a function that outputs an image given some representation of that image (an *image code*) as input. We tested the family of DeePSiM generators developed in [8]; they are generative adversarial networks trained to invert each layer of AlexNet [23]. The pretrained models are available at https://lmb.informatik.uni-freiburg.de/people/dosovits/code.html. We have converted the models into PyTorch for convenience for future research. Links to the converted models are available in the code repository (see Code availability below). We used the image generator inverting the fc6 layer by default except in S4 Fig, where we compared different generators. An alternative version of the DeePSiM-fc6 generator was trained on the Places-365 dataset using code from [8] and a pre-trained classifier [24].

### Fitness function

The key metric XDream optimizes is a scalar value we refer to as *fitness*, which is associated with each image. In the neuroscience context, a fitness function can be the stimulus-evoked spike count for a neuron in visual cortex. In the current study, the fitness function is the activation of the target unit in a ConvNet.

### Optimization algorithms

An optimization algorithm in the context of XDream is a function that iteratively proposes a set of $n$ image codes (real-valued vectors) or *codes* for short, $c_i$, $i = 1, \ldots, n$, and then uses their corresponding *fitness* values $y_i$, $i = 1, \ldots, n$ to propose a new set of codes expected to have higher fitness. We used a genetic algorithm by default, but also considered two other algorithms: finite-difference gradient descent (FDGD) and natural evolution strategies [25] (NES). Implementation details for the optimization algorithms are available in S1 Text.

### Computing environment

Neural network computations were performed on NVIDIA GPUs. Portions of this research were conducted on the O2 High Performance Compute Cluster supported by the Research Computing Group, at Harvard Medical School. See http://rc.hms.harvard.edu for more information.

## Results

### Random exploration of stimulus space is inefficient

A common approach for exploring neuronal selectivity is to use arbitrarily selected images, often from a limited number of categories (for example in [7, 26]). Thus, we considered random exploration as a baseline for comparison. We used the AlexNet architecture as the target model [23] (implemented as CaffeNet; S1 Table) and sampled images from ImageNet [27] (ILSVRC12 dataset, 1,431,167 images), a large dataset common in computer vision that also contains the training set of CaffeNet. We randomly sampled $n$ images either from all of ImageNet or from 10 categories randomly selected from the 1,000 training categories in ImageNet ($n/10$ images per category). For units in different layers of the network, we evaluated the activation values in response to these images and calculated the *relative activation*, defined as the ratio between the activations in the $n$ random images and the maximum activation in all of ImageNet. By definition, the relative activation for the best image in ImageNet is 1, which is also an upper bound on the observed relative activation values when using random sampling. Randomly selected images typically yielded relative activation values well below 1 (S1 Fig). As expected, the maximum observed relative activation increased with $n$ but only did so slowly, with near-logarithmic growth. Moreover, for later layers (e.g., fc8), sampling from only 10 categories yielded significantly worse results than sampling completely randomly, which we hypothesize is because the small number of categories imposes a bottleneck on the diversity of high-level features represented. In neuroscience studies, category selection is clearly not completely random: Investigators may have intuitions and prior knowledge about the types of stimuli that are more likely to be effective. To the extent that those intuitions are correct, they can enhance the search process. However, those intuitions are seldom guided by systematic examination of stimulus space and could well miss important types of stimuli.

## XDream can find strong stimuli for neurons

XDream has three key components: an image generator representing the search space; an objective function given by the activation of the target unit guiding the search; and an optimization algorithm performing the search (Fig 1a). In each generation, the generator creates images from their latent representations (*codes*), the target unit activation is evaluated for each of the generated images, and the optimizer refines the codes based on the activation values. Initialized randomly (examples shown in Fig 1a), the algorithm is iterated for 10,000 total image presentations, a relatively small and accessible number in a typical neuroscience experiment [13]. Crucially, the algorithm does not use any prior knowledge about the architecture or weights of the target model.

An example experiment with unit 1 in the output layer (layer fc8) of CaffeNet is shown in Fig 1b and 1c. In 500 generations of 20 images each, the activation of the target unit increased rapidly and saturated at approximately generation 300. Fig 1c shows example images at a few generations (log-spaced to show a range of activations), illustrating the evolution of the images from the initial noise pattern to the final image. In the following analyses, we concentrate on the best image in the last generation, which we refer to as the *optimized image*. However, it is worth noting that responses to all the 10,000 unique images during the evolution may illuminate features of the neuron's tuning (see Discussion).

How strong was the activation achieved by XDream-generated images? We compared the optimized image to images from ImageNet. Unit 1 in layer fc8 was trained to be a "goldfish" detector. Correspondingly, when we randomly sampled 10,000 images from ImageNet, the best images are photos of goldfish (Fig 1d). The highest activation value observed in this random sample was 30.67. The best image from ImageNet for this unit was a picture of a goldfish and elicited an activation of 40.55 (Fig 2b). Consistent with S1 Fig, the best image found by random sampling produced a much lower activation value than the best example in ImageNet. In comparison, the optimized image generated by XDream elicited an activation of 72.42. In other words, using a limited number of presentations, XDream generated images that elicited higher activation than any natural image from ImageNet. We refer to such images with relative activation $> 1$ as *super stimuli*.

## XDream generalizes across layers, architectures, and training sets

The default generative network used in XDream was trained to invert the internal representations at layer fc6 of CaffeNet, which was in turn trained on ImageNet [8]. Could this generator allow XDream to generalize to other network layers, architectures, and training sets? If XDream is specific to certain layers and architectures, or specific to ImageNet-trained networks, this may limit its applicability to real neurons.

We first assessed whether XDream could extrapolate to other layers in CaffeNet by selecting 100 units respectively from the early, middle, late, and output layers of CaffeNet (Fig 2a). XDream was able to find optimized images that are better than the best randomly selected images across all layers ($p < 10^{-16}$, false discovery rate (FDR) corrected for 28 tests in this section). The optimized images were also significantly better than the best images in ImageNet ($p < 10^{-9}$, FDR corrected).

Next, we tested 100 units from each of 4 layers from 5 different network architectures: ResNet-v2 152- and 269-layer variants [28], Inception-v3 [29], Inception-v4, and Inception-ResNet-v2 [30]. These models were all trained on ImageNet. XDream was able to generate better images than the best random images for the vast majority of units across all layers and architectures (Fig 2a; $p < 10^{-8}$ across layers) except the early layer of Inception-v3 ($p = 0.2$) and of Inception-ResNet-v2 ($p = 0.09$). With the same exceptions, XDream generated super

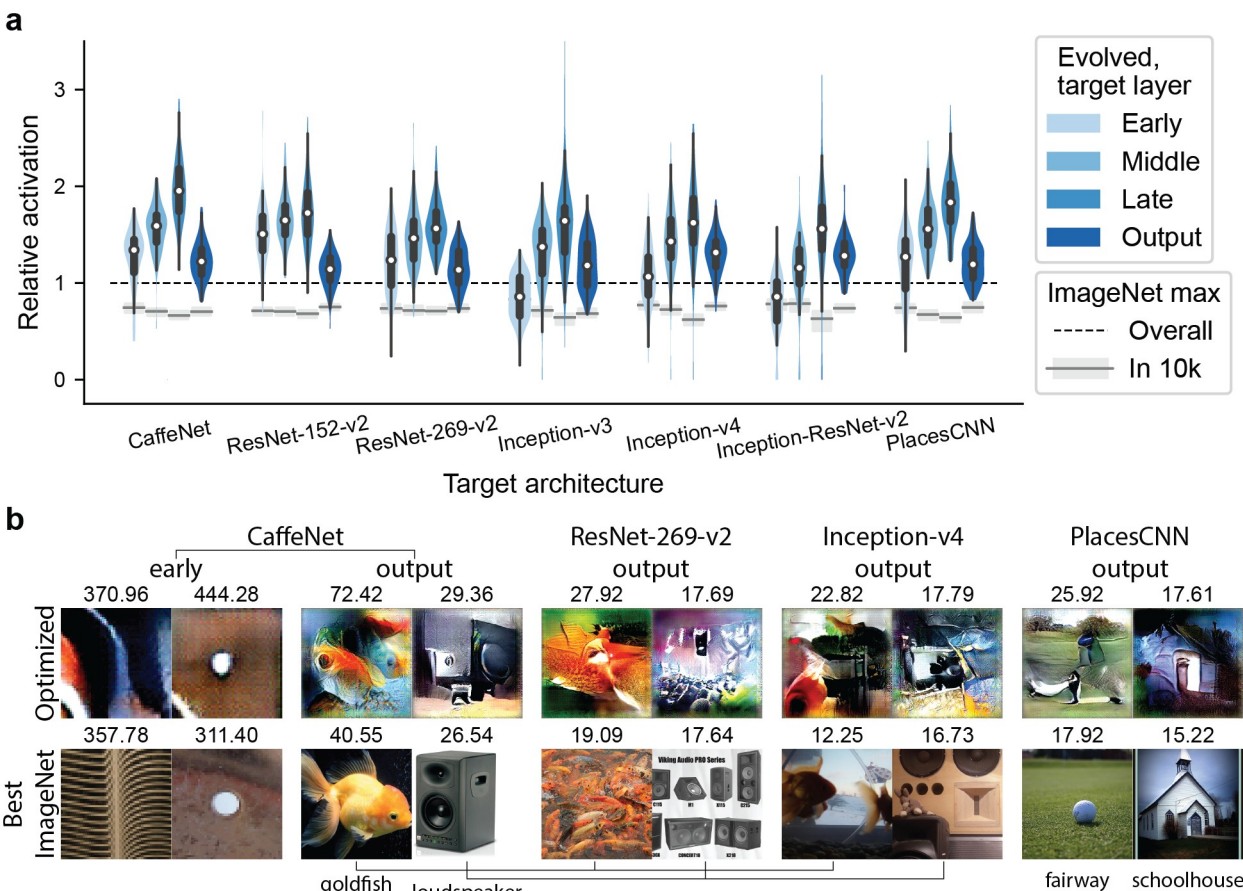

**Fig 2. XDream generalizes across layers, architectures, and training sets. a)**, Violin plot showing the distributions of relative activation (activation of optimized stimulus relative to highest activation in >1.4 M ImageNet images) over 100 randomly selected units per layer. For each target model, we investigated early, middle, late, and output layers (see S1 Table for the specific layers). The violin contours indicate kernel density estimates of the distributions, white circles indicate the medians, thick bars indicate first and third quartiles, and whiskers indicate 1.5× interquartile ranges. For comparison, grey boxes (interquartile ranges) and lines (medians) show the distribution of maximum relative activation for 10,000 random ImageNet images. The horizontal dashed line corresponds to the best ImageNet image. **b)**, Optimized (top row) and best ImageNet (bottom row) images and activations for 10 example units across layers and architectures. For output units, corresponding category labels are shown below the images.

stimuli for all other tested layers ($p = 0.01$ for the early layer of Inception-v4, $p = 2 \times 10^{-4}$ for the middle layer of Inception-ResNet-v2, and $p < 10^{-9}$ for all other layers). Example optimized images for units in different layers and architectures are shown in Fig 2b and S2 Fig. Furthermore, several generators trained on different layer representations performed equally well across classifier layers (S4 Fig).

Finally, we tested the ability of XDream to optimize unit responses when the generator and target networks are trained on different datasets. We tested PlacesCNN [31], a network with the same architecture as CaffeNet but trained on a different dataset, PlacesCNN. PlacesCNN also contains photographic images, but they mainly depict scenes rather than objects. Again, XDream was able to find super stimuli across all layers in this network (Fig 2a, last four distributions; $p < 10^{-6}$ across layers), even when using a generative network trained on different images. Conversely, when using a generator trained on the Places dataset, XDream still performed similarly well in optimizing CaffeNet and PlacesCNN (S4 Fig).

These results show that XDream can efficiently create images that trigger high activations in a target unit without making assumptions about the type of images a unit may prefer and without any knowledge of the target model architecture or connectivity, suggesting that XDream may well be applicable to biological neurons. Furthermore, XDream generalizes across layers in a ConvNet, while different layers roughly correspond to areas along the ventral visual stream [17, 32, 33], suggesting that XDream may also generalize to several ventral stream areas. Consistent with this observation, results from [13] indicated that XDream can find optimized stimuli for V1 as well as inferior temporal cortex (IT) neurons.

## XDream is robust to different initial conditions

XDream starts the search from an initial generation of image codes. In Fig 2, we always initialized the algorithm using the same set of 20 random image codes, 6 of which are shown in Fig 1a. Does the choice of initial conditions affect the results?

To address this question, we first tested how much the particular choice of random initial codes matters. For each target unit, we repeated the experiment using 10 different random initializations and compared the optimized relative activation to that of the original random initialization. Different initial conditions produced slightly better or worse relative activation values centered around a mean difference of 0, and the standard deviation of the fractional change was lower than 10% (Fig 3a).

Similar activation values notwithstanding, the optimized images were different on a pixel level (Fig 3b); they may comprise an "invariance manifold" that contains similar, but not identical, images eliciting comparable activation values. What might this invariance manifold look like? To explore this question, in CaffeNet layers conv2, conv4, fc6, and fc8, we linearly interpolated between two separately optimized images (from different initializations) in the image code space, and measured target unit activation in response to the interpolated images (Fig 3b). The interpolated images were much stronger stimuli compared to the majority of ImageNet images. However, particularly in layers fc6 and fc8, the interpolation midpoint activated the units less strongly than either endpoint, suggesting either that the sets of strong stimuli are disjoint, or that the invariance manifolds may have non-convex geometry. Studies have reported visual neurons that prefer seemingly unrelated stimuli. It remains an interesting open question to identify whether there exists a feature representation space in which neuronal tuning functions have "simple" geometry.

Next, we tested whether there are particularly good or bad ways of choosing the initial stimuli. We selected, separately for each target unit, the 20 ImageNet images that led to the highest, middle, and lowest activation values and used those images to form the initial population (Fig 3d). To convert images into image codes comprising the initial population, we used either the "opt" or the "ivt" algorithm (Methods). Initializing with better or worse natural images did not improve the optimized images in the conv2 layer ($p = 0.87$ and $0.19$ for "opt" and "ivt," respectively, FDR-corrected for 8 tests in this and the next sentence). In higher layers, initializing with the best natural images led to slightly higher relative activation values (Fig 3d; Table 1; $p < 5 \times 10^{-3}$ for "opt" and $p < 10^{-10}$ for "ivt" across layers). We speculate that the improvement in higher layers is because units in deeper layers are progressively more selective, making it more difficult to optimize their responses. Therefore, more optimal initializations are beneficial. However, in an actual neurophysiology experiment, it is unlikely that the investigator would know, *a priori*, such good stimuli as the best of 1.4 M images. Meanwhile, initializing with the middle or worst natural stimuli were similar to initializing with random images codes. Therefore, initializing randomly seems reasonable.

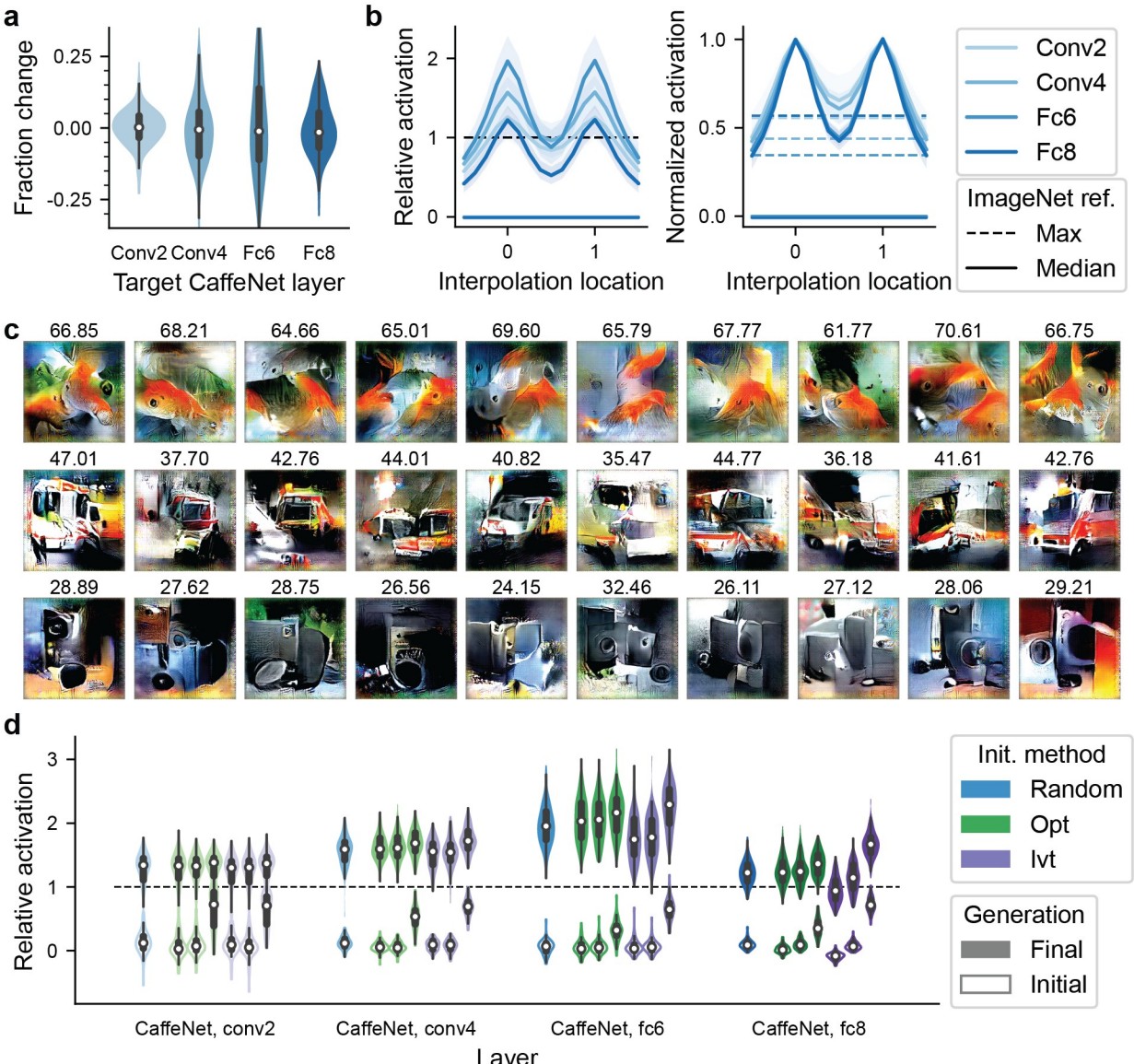

**Fig 3. Comparison of different initializations. a,b,c),** Effect of using different random initializations. **a),** Distributions of fractional change in optimized activation if 10 different random initializations are used. **b),** Left, relative activation in response to images interpolated (in the code space) between two optimized images from two different random initial conditions. Right, activation normalized to the endpoints (location 0 or 1), highlighting the change in activation away from the endpoints. **c),** Optimized images from different initializations for 3 example units in the output layer (one unit per row). Activation values are shown above each image. **d),** Good versus bad initializations. For each target unit, its best, middle, or worst 20 images from ImageNet were used as the initial generation. The images were converted to the image code space using either an optimization method ("opt") or an inversion method ("ivt"; Methods). Left to right within the opt and ivt groups are results from initialization with the worst, middle, and best 20 images. Random initialization is shown for comparison. The open and solid violins show the distributions, in the first and last generation respectively, of relative activation over 100 units in each layer.

To summarize, initializing the algorithm with different random conditions resulted in only a small variation in the optimized image activation, and the optimized images were similar, although not identical, at the pixel level. Initializing with prior knowledge has little to no effect on the optimized image activation, unless the seed is comparable to the best image in $\sim 1\,\mathrm{M}$ images and only in later layers.

**Table 1. Effect of using good vs. bad initialization.**

| Encoding alg. | Measure | Layer | | | |
|---|---|---|---|---|---|
| | | **conv2** | **conv4** | **fc6** | **fc8** |
| opt | slope | 0.010 | 0.037 | 0.047 | 0.056 |
| | p-value | 0.87 | 0.004 | 0.004 | $7 \times 10^{-5}$ |
| ivt | slope | 0.044 | 0.113 | 0.241 | 0.353 |
| | p-value | 0.19 | $7 \times 10^{-11}$ | $4 \times 10^{-22}$ | $5 \times 10^{-77}$ |

For each unit, the 20 worst, middle, and best images from ImageNet, as ranked by that unit, were used to initialize the genetic algorithm. The images were converted to image codes using one of two encoding algorithms, "opt" or "ivt" (see Methods). The slope was calculated, by linear regression, for relative activation (median across 100 random units each layer) as a function of the initialization ({0,1,2} for {worst, middle, best}, respectively). Thus, the slope quantifies the improvement in relative activation when a better initialization is used (worst → middle or middle → best).

## Different optimization algorithms can be incorporated into XDream, but the genetic algorithm consistently works well

An important component of XDream is the optimization algorithm. The results shown thus far were based on using a genetic algorithm as the optimization algorithm, a choice inspired by previous work [9–11]. Here, we compared the genetic algorithm to two additional algorithms, a naïve finite-difference gradient descent algorithm (FDGD; Methods) and Natural Evolution Strategies (NES; [25], Methods). NES has been used in a related problem [34]. FDGD and NES were significantly worse than the genetic algorithm in CaffeNet conv2 ($p < 10^{-13}$, FDR corrected for 20 tests here and in the next section) and conv4 layers ($p < 10^{-3}$). Yet, both FDGD and NES were significantly better than the genetic algorithm in CaffeNet fc6 ($p < 10^{-16}$), fc8 ($p < 10^{-16}$), and Inception-ResNet-v2 classifier layers ($p < 10^{-12}$; Fig 4a).

## XDream is robust to noise in neuronal responses

An important difference between model units and real neurons is the lack of noise in model unit activations. Upon presenting the same image, a model unit returns a deterministic activation value. In contrast, in biological neurons, the same image can evoke different responses on repeated presentations (even though trial-averaged responses may be highly consistent; see [35]). To test whether XDream could still find super stimuli with noisy units, we implemented a simple model of stochasticity in the units by using the true activation value to control the rate of a homogeneous Poisson process, from which the "observed" activation value on a single trial was drawn (Methods). Homogeneous Poisson processes have been used extensively to model stochasticity in cortical neurons [36].

As expected, performance deteriorated when noise was added (Fig 4a, noisy condition). However, XDream using the genetic algorithm was still able to find optimized stimuli better than random exploration for most layers ($p < 10^{-10}$ for all tested layers except $p = 0.19$ for CaffeNet fc8, FDR-corrected for 5 tests) and was also able to find super stimuli for some layers ($p < 10^{-5}$ for CaffeNet conv4 and fc6 layers; FDR-corrected for 5 tests).

Noise in the unit activations affected different optimization algorithms to different extents. The genetic algorithm was at least as good as, and often superior to, both alternative optimization algorithms when considering noisy units. The NES algorithm performed similarly to the genetic algorithm in CaffeNet fc8 layer and Inception-ResNet-v2 classifier layer ($p = 0.03$ and 0.65, respectively), but was worse in the other 3 tested layers ($p < 10^{-14}$). The FDGD algorithm

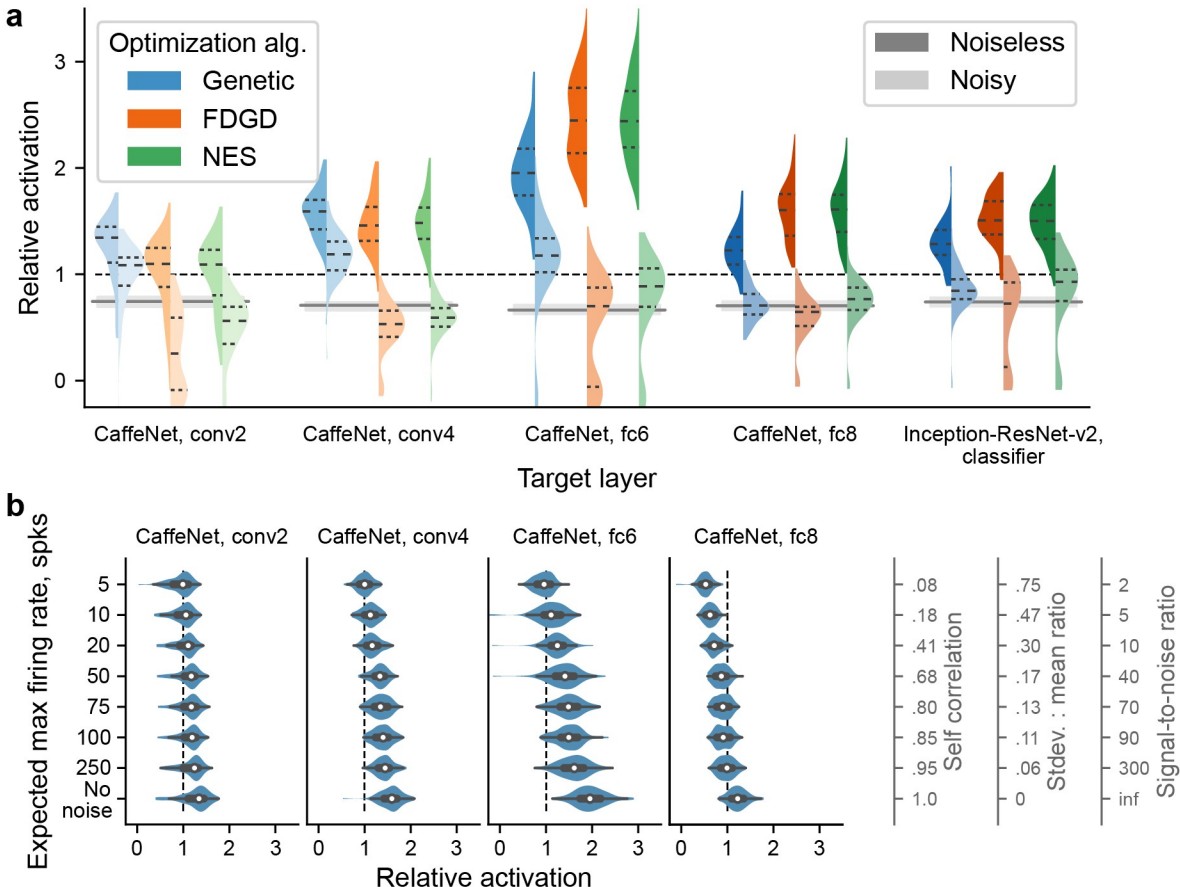

**Fig 4. Comparison of optimization algorithms and their robustness to noise.** We compared 3 gradient-free optimization algorithms (Methods): a genetic algorithm, finite-difference gradient descent (FDGD), and Natural Evolution Strategies (NES; [25]). Left and right half of each violin, respectively, correspond to noiseless and noisy units. Dashed lines inside the violins indicate quartiles of the distribution. Otherwise, format of the plot is as in Fig 2a. **b)**, The performance of the genetic algorithm gradually improves with decreasing amounts of noise within a neurophysiologically relevant range. Format of the plot is as in Fig 2a except that the violins are horizontal. On the right, 3 alternative scales for the y-axis are shown, for comparison with common ways of assessing noise.

was particularly sensitive to noise, performing worse than the genetic algorithm in all layers tested ($p < 10^{-6}$) and frequently failing to find good stimuli.

In the noisy conditions examined thus far, we assumed that in each presentation, model units yielded approximately 20 spikes for a "good" stimulus (defined as the expected best image in 2,500 random ImageNet images). This choice was motivated by what may be realistically expected when recording from biological neurons (e.g., firing rate of 100 spikes per second to a good stimulus over a 200 ms observation window), but this number will be dependent on individual neurons and specific experimental designs. This number matters because, for a homogeneous Poisson process, its standard deviation-to-mean ratio is inversely proportional to the square root of the rate parameter (average expected number of spikes), and thus a higher firing rate means a higher signal-to-noise ratio. To characterize the performance of XDream under different noise conditions, we varied the rate parameter as defined by the expected max spike number and measured XDream performance on the different noise levels (Fig 4b). The empirical level of noise was quantified with commonly used measures such as trial-to-trial self-correlation, standard deviation-to-mean ratio, and signal-to-noise ratio (SNR). As the amount of noise decreased, the performance of XDream gradually approached

its noiseless performance. Notably, even with a high level of noise (5 spikes for a good stimulus, self-correlation of 0.08, and SNR of 2), XDream was able to find super stimuli for around half of the target units in all but the deepest layer (fc8) tested.

## Availability and future directions

The code for XDream can be obtained directly from https://github.com/willwx/XDream/.

In the computer science literature, activation maximization is a well-known approach for visualizing features represented by units in a ConvNet [12, 21, 37–39]. However, the techniques are only applicable to networks that provide optimization gradients. In other words, perfect knowledge is assumed of the target network architecture and weights. Clearly, such requirements are not met in current neuroscience experiments.

Recently, several other studies have focused on similar goals to the ones in XDream, but with a different approach [32, 33, 40, 41]. In that approach, a ConvNet-based model is first fitted to predict neuronal responses to a set of training images. Then, standard white-box activation maximization techniques are applied to the ConvNet model. The relation between this approach and XDream is similar to the relation between the so-called "substitute model" approach and what, in comparison, we may call a "direct" approach, in research on black-box adversarial attacks. A promising future direction is to combine the two approaches to leverage their unique advantages: unlimited queries (after training) and efficient optimization with substitute models; avoiding model extrapolation and transferability problems with direct optimization.

The results presented here are based on maximizing activation values, whereas the results shown in [13] are based on maximizing spike counts. Activation values and firing rates are commonly-used proxies for internal representation in machine learning and neuroscience, respectively. However, other putative neural codes can be studied, such as pooled activation across multiple units, increase sparseness of the representation across units, match a pre-specified pattern of population firing, correlated firing, synchronized firing of nearby units, maximize power in a certain frequency band in local field potentials, etc. XDream is agnostic to the underpinning of the objective function as long as it is image-specific, quantitatively defined, and computable in real time. Thus, the same algorithm can be readily applied to investigate different putative neural coding mechanisms.

Finally, it is worth remembering that the identification of an optimal stimulus, or even a diverse set of them, still does not automatically lead to a full characterization of the function of a neuron. Finding preferred stimuli, or "feature visualization" in computer science parlance, has guided thinking about the function of individual neurons in both neuroscience and deep learning [6, 21]. However, optimal stimuli reflect but do not disentangle critical issues like tuning features, invariant features, and context dependence; these questions need to be distinguished by subsequent hypothesis-driven investigation [42–44]. A method to automatically find preferred stimuli of neurons can suggest initial hypotheses about a poorly-understood visual area, or motivate re-thinking about an extensively-studied region. Of note, during the optimization process, XDream does test thousands of related images, covering the target unit's response levels both widely and densely (Fig 1c). Closer analyses of these images may reveal richer information about the tuning surface of a neuron (e.g., invariances) than what is reflected by the single best image.

In summary, XDream is able to discover preferred features of visual units without assuming any knowledge about the structure or connectivity of the system under study. Thus, XDream can be a powerful tool for elucidating the tuning properties of neurons in a variety of visual areas in different species, even where there is no prior knowledge about the neuronal

preferences. Furthermore, we speculate that the general framework of XDream can be extended to other sensory domains, such as sounds, language, and music, as long as good generative networks can be built.

## Supporting information

**S1 Fig. Expected maximum relative activation in response to random natural images.** We measured the max relative activation expected in two random sampling schemes. "Random" refers to picking a given number of images randomly from the ImageNet dataset (blue). "10 categories" refers to first randomly picking 10 categories out of the 1000 ImageNet categories and then picking randomly from those categories (gray). We considered 4 layers from the CaffeNet architecture. Lines indicate the median relative activation (activation divided by the highest activation for all ImageNet images). Shading indicates the 25th- to 75th-percentiles among 100 random units per layer.
(TIF)

**S2 Fig. Optimized and best ImageNet images for other example neurons across architectures and layers.** Two neurons were randomly selected per layer per architecture (S1 Table). Format is the same as in Fig 2.
(TIF)

**S3 Fig. The image generator can approximate arbitrary images, and XDream can find these images using only scalar distance as a loss function.** This figure reproduces Supplementary Figure 1 in [13]. The generative network is challenged to synthesize arbitrary target images (row 1) using one of two encoding methods, "opt" (row 2) and "ivt" (row 3; Methods). In addition, XDream can discover the target image efficiently (within 10,000 test image presentations) by using the genetic algorithm to minimize the mean squared difference between the target image and any test image as a loss function, either in pixel space (row 4) or in CaffeNet pool5 representation space (row 5).
(TIF)

**S4 Fig. Comparison of image generators. a)** We tested each of the family of image generators from [8] as the image generator in XDream, together with a generator directly representing images as pixels. Format of the plot is the same as in Fig 2a. **b)**, The same generator architecture (DeePSiM-fc6) was trained on ImageNet and Places365, respectively, and tested on classifiers trained on either dataset. Each half of a violin corresponds to one generator, and dashed lines inside the violins indicate quartiles of the distribution; otherwise, format of the plot is the same as in Fig 2a.
(TIF)

**S5 Fig. Comparison of hyperparameters in the genetic algorithm.** In each plot, one hyperparameter was varied while the others were held constant at default values indicated by the open circles. Dots indicate the mean of relative activation across 40 target neurons, 10 neurons each in 4 layers specified in S4 Table. Blue and orange lines indicate noiseless and noisy target units, respectively. Light colored lines indicate the mean across the 10 units within each architecture and layer. Light gray shading indicates the linear portion of a symmetrical log plot, which is used in order to show zero values.
(TIF)

**S6 Fig. Testing XDream on a toy model that mimics the extra-classical effect of surround suppression.** We took two feature channels (first column, rows 2 & 3) from the conv1 layer of AlexNet and tiled each spatially with positive and negative weights to create a central, circular

excitatory region and a concentric suppressive ring, analogous to an excitatory classical receptive field (RF) and a suppressive extraclassical RF (first row). By maximizing responses of the constructed units, XDream created stimuli that are spatially confined and agreed with the varying RF sizes (rows 2 & 3). We also created a unit that preferred a horizontal pattern in the center and a vertical pattern in the surround; XDream was able to uncover this preference pattern as well (row 4).
(TIF)

**S1 Table. Target networks and layers.** For each network, 4 layers from what is roughly the early, middle, late stages of processing, together with the output layer before softmax, were selected as targets. PlacesCNN has the same architecture as CaffeNet but is trained on the Places-205 dataset [31]. CaffeNet is as implemented in https://github.com/BVLC/caffe/tree/master/models/bvlc_reference_caffenet, PlacesCNN as in [31], and the remaining as in https://github.com/GeekLiB/caffe-model.
(PDF)

**S2 Table. Optimized hyperparameter values for the genetic algorithm.** Hyperparameters used in the experiments in this paper, obtained as described in Methods separately for each generative network and for noiseless and noisy targets.
(PDF)

**S3 Table. Optimized hyperparameter values for the FDGD and NES algorithms.** Hyperparameters used in the experiments in this paper, obtained as described in Methods separately for the noiseless and noisy case. The generative network was always deepsim-fc6.
(PDF)

**S4 Table. Inferior temporal cortex-like layers.** From each layer, 10 units were randomly selected and used in hyperparameter evaluation.
(PDF)

**S1 Text. Methods and additional experiments & discussion.**
(PDF)

## Author Contributions

**Conceptualization:** Will Xiao, Gabriel Kreiman.

**Data curation:** Will Xiao.

**Formal analysis:** Will Xiao.

**Funding acquisition:** Gabriel Kreiman.

**Investigation:** Will Xiao.

**Methodology:** Will Xiao, Gabriel Kreiman.

**Project administration:** Gabriel Kreiman.

**Resources:** Gabriel Kreiman.

**Software:** Will Xiao.

**Supervision:** Gabriel Kreiman.

**Visualization:** Will Xiao, Gabriel Kreiman.

**Writing – original draft:** Will Xiao, Gabriel Kreiman.

**Writing – review & editing:** Will Xiao, Gabriel Kreiman.

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
