## [Decision Letter · Decision Letter 0]

25 Jan 2020

Dear Mr. Xiao,

Thank you very much for submitting your manuscript "XDream: finding preferred stimuli for visual neurons using generative networks and gradient-free optimization" for consideration at PLOS Computational Biology.

As with all papers reviewed by the journal, your manuscript was reviewed by members of the editorial board and by several independent reviewers. In light of the constructive reviews (below this email), we would like to invite the resubmission of a significantly-revised version that takes into account the reviewers' comments.

We cannot make any decision about publication until we have seen the revised manuscript and your response to the reviewers' comments. Your revised manuscript is also likely to be sent to reviewers for further evaluation.

Thank you again for your submission. We apologize for the length of time these reviews took, but I can assure you we were working to obtain reviews for the entire time your paper was under review.  We hope that our editorial process has been constructive so far, and we welcome your feedback at any time. Please don't hesitate to contact us if you have any questions or comments.

Sincerely,

Alona Fyshe, Ph.D.

Associate Editor

PLOS Computational Biology

Wolfgang Einhäuser

Deputy Editor

PLOS Computational Biology

Reviewer's Responses to Questions

**Comments to the Authors:**

Reviewer #1: The authors produce an analysis of the XDream method that uses generative networks with a genetic algorithm for obtaining stimuli that will optimally drive neurons. They test this method by using it on pre-trained deep neural networks. They show that XDream can consistently find stimuli that will drive units in pre-trained networks to responses greater than those of the image set from which the networks were trained. This result is approximately independent of depth in the network and is independent of initial conditions. It is relatively robust to the choice of generative network, as long as that generator produces a "high-level" representation. These results are somewhat robust to the choice of optimization but are impacted by adding noise to the representation.

This looks at an important issue for visual neuroscience, that of finding stimuli that drive neurons in a manner that is not plagued by various experimental and theoretical biases (cf. Olshausen-Field "What is the other 85% of V1 doing?" and Carandini, et al 2005, "Do we know what the early visual system does?"). The XDream tool is an example of an emerging methodology for dealing with this problem and studying its capabilities and drawbacks is therefore important and useful to the field. The paper is clear. I recommend publication after the authors have addressed the feedback below.

1) The most important issue the authors should address is to provide some acknowledgement or discussion of the notion of optimal or "preferred" stimuli in the first place. By their methodology, they are looking for a single image that will optimally drive neurons, but this is probably not the right way to think about the coding problem, certainly not for neurons and probably not for the units in deep networks as well. The "single best image" approach is going to gloss over issues like contextual dependence. These optimal stimuli may in fact be a very small set that drives responses because they combine some "feature" for which the neuron/unit codes with exactly the right context to enhance the response. Depending on the size of this space and the nature of the context, one might wind up with an image that is not very informative of the actual coding properties of the neuron or unit. The authors use the phrase "true feature preference" in the introduction, but need to acknowledge that their method may not find it either. Furthermore, this problem is compounded in the case of real neurons, in which there are feedback and lateral inputs that produce extra-classical effects. These effects may or may not appear in the models that the authors are testing, since they are all feedforward networks. This set of questions will almost certainly lead to the "invariance manifold" that the authors discussed.

None of this takes away from the authors' work, but users of the method should be aware of these issues.

2) Related to point 1, the authors should comment on the possible ethological relevance of super-stimuli. It is not surprising that such stimuli exist. Given that each unit produces a 1D parameterized function of image space, one should be able to find a point in the input space that produces a more extreme result than any finite set of inputs. Do these have a useful meaning in terms of describing that function?

3) The authors showed that the generative model in XDream was "expressive", but those demonstrations also show what the model does not capture, namely high-frequency content of images (fine details are lost in all the examples in (what should be) Figure 3). Is this a limitation for the methodology? It seems so, since the model shouldn't then be able to capture optimal stimuli for neurons that respond to fine details. I suspect this is not a severe practical limitation, but it is there.

4) The authors should acknowledge the bias that is built into their methodology by looking at samples from fixed image databases. ImageNet does have a particular structure and, unless I'm mistaken, all the tested generators were trained on this structure (implicitly by inverting AlexNet). This will bias towards finding features similar to those necessary for describing ImageNet in particular. This is not just an issue of whether the generative model can reproduce an image when forced to (e.g. the Figure that should be Figure 3) but whether it will tend to promote certain features over others. ImageNet will provide an implicit bias.

5) The claim about robustness to noise is overstated. Figure 6 makes the technique look quite susceptible to Poisson noise, depending upon target layer.

Minor issues:

1) Figure 1b and 1c: there are 9 minor tick marks and ten example images.

2) Figure 2 and 3 appear to be swapped (at least in the pdf I received).

3) Figure 2: label the random ImageNet samples in the Figure (grey boxes). label the dotted line as "maximum ImageNet response" or something similar. This will make the figure easier to read and digest.

4) line 112: ...we qualitative[ly] assessed...

5) lines 136-7: delete either "it can generate" or "can be generated"

6) Figure 4: label the open and solid violin plots on the Figure

7) Table 1: the caption appears to be a Latinate placeholder.

8) lines 211-9: this section needs to be rewritten. There needs to be a reference to Table 1, otherwise "slope" is introduced before the reader has seen such a thing. It isn't in a figure or preceding discussion. One is left to infer what variables the linear regression is being performed upon.

Reviewer #2: I will upload the pdf with my comments in it, where I have found typos and made wording suggestions.

Review of XDream: finding preferred stimuli for visual neurons using generative networks and gradient-free optimization (Will Xiao and Gabriel Kreiman)

by Gary Cottrell

This paper uses an existing method for finding optimal visual stimuli for neurons that has previously been applied in Macaque in a paper in Cell in 2019. The goal here is to elucidate how robust the method is by applying it to several deep network models with varying architectures and at different layers of the networks. Using network models allows for extensive experimentation that is impractical in biological preparations. The model proves to be very robust to various parameter regimes, and finds stimuli that drive the neuron much more than any of the over 1 million images in the ImageNet dataset. One of the most interesting findings here is that, using different image generators or different initial conditions, the model finds multiple images that drive the neurons similarly, and these images resemble one another to the human eye. Hence they suggest that there is an optimal image manifold in the latent space of images. Unfortunately, this point was already made in the prior paper with actual monkey visual neurons.

Since the authors postulate that there is an invariance manifold, it would be useful to test this idea by looking at what is generated by a linear interpolation of the codes found from different initial conditions, and how well those interpolated images drive the model neuron. While it is unlikely that the manifold is linear, since the images are similar, they are probably nearby in this space.

In the paragraph on other things one could study with this approach, such as correlated firing, synchronized firing, LFPs, etc., it would be helpful to say what you would optimize in a couple of cases. E.g., you might say, “for example, we could optimize based on increased correlations in firing rates between neurons” or some such.

The paper could be improved by moving more of the information into the methods section or into supplementary material. There are many details that make the exposition rather tough sledding for the reader. In particular, the section on the effects of different generators has a couple of caveats, e.g., “except for CaffeNet conv2”, and “The pixel-based image generator, compared to generative neural networks, worked more poorly in all target layers other than CaffeNet conv2 except when compared to deepsim-norm2 (p = 1 compared to deepsim-norm2; p > 0.14 compared to other generators in CaffeNet conv2; p < 10^−4 in all other comparisons; FDR-corrected for 32 tests comparing each generator to raw-pixel in each target layer).” This amount of detail makes my eyes glaze over. I think this section doesn’t add a lot to the paper, and could profitably be moved to the supplementary material.

Similarly, the section on different optimizers doesn’t flow well. The GA works better on some layers and the other two algorithms work better in some other layers. I’m not sure I care, and I’m not sure what the take-home message is. Again, there are similar “this works better in this layer and that works better in other layers” results in the noise experiments. I think the noise experiments are important, as this is a more realistic case. Unfortunately, the results in Figure 6 are not encouraging. I would disagree with the header for this section: “XDream is robust to noise in neuronal responses.” It doesn’t seem that robust. I wonder if there is a fix for this - for example, can you average the rate over some time interval of the Poisson process instead of simply sampling from it? Unfortunately, I don’t know enough about Poisson processes to know if this is a good suggestion or not.

One concern is that, although the authors state that “we focus on the more biologically relevant scenario where there is no information about the architecture and weights of the target model, and where we only have access to a few, potentially stochastic, activation values from the neurons.” In fact, they don’t have access to only a few activations. The model is used to generate 10,000 images to find the optimal one. This seems biologically unrealistic. I skimmed the previous paper, and there they used many fewer generations for the monkeys - 200. Since this is what is apparently possible in biological preparations, it seems like they should evaluate how much is gained in 200 generations and compare that to the 500 generations they used here. This would provide a better estimate of what is possible today.

Originality

The originality is tempered somewhat by the fact that this is investigating an existing method that has already been used in Macaque visual cortex. Obviously, this is different in the sense that using convnets as the preparation allows for much more extensive experimentation with the method than would be possible with a biological one, which is the point of this paper. For example, they can test the neuron’s response to the over one million images in ImageNet, and then compare the results of this brute force search to the effectiveness of XDream. They vary just about everything and still find that the method works well. So it is original in that sense, but I have to think that the big-font headline is the first paper.

Innovation

Again, the innovation here is to benchmark their method using in silico models. This kind of thing has been done before with other analyses of deep net features.

High importance to researchers in the field 

To the extent to which other researchers may start to use this method, the importance here is that neuroscientists can get some assurance that the method is robust, and that they needn’t worry too much about optimizing the meta-parameters.

“Similar activation values notwithstanding, the optimized images were different on a pixel level (Fig 4b); they may comprise an ”invariance manifold” for each neuron that contains similar but not identical images eliciting comparable activation values (see Discussion).” I think this is an important point, so it would be good to test the hypothesis by looking along a line between some of the image codes to see if there are optimal images between these. This would be of interest to researchers using this method.

Significant biological and/or methodological insight 

There is not much in the way of biological insight here, but the paper does demonstrate that this methodology is robust.

Rigorous methodology

This is quite a rigorous test of the model.

Substantial evidence for its conclusions

The evidence that this approach works well over a variety of convnet architectures and layers is extensive. On the other hand, I think the evidence for robustness to noise is weak. This is a point the authors should address in a revision of the paper.

Minor comments:

I’m not quite sure what this sentence means: “the standard deviation was lower than 10% of the activation values (Fig 4a).”

I’m not sure the slopes in Table 1 give a very intuitive idea of how much improvement you get by using good vs. bad initializations. A bar chart might be better. Also, while the text suggests that there isn’t much difference, the slopes seem fairly big as you go deeper in the net using the ivt method. I don’t know if that conclusion of mine is warranted. I.e., I would better understand this if the actual differences in medians were shown instead of the slope. Perhaps another violin plot would work here? It seems slightly counterintuitive to conclude that there is only slight variation in the optimized image activation while the p-values for the differences are on the order of 10^-77.

**Have all data underlying the figures and results presented in the manuscript been provided?**

Reviewer #1: None

Reviewer #2: None

PLOS authors have the option to publish the peer review history of their article (what does this mean?). If published, this will include your full peer review and any attached files.

Reviewer #1: No

Reviewer #2: Yes: Garrison W Cottrell
---

## [Decision Letter · Decision Letter 1]

12 May 2020

Dear Mr. Xiao,

Thank you very much for submitting your manuscript "XDream: finding preferred stimuli for visual neurons using generative networks and gradient-free optimization" for consideration at PLOS Computational Biology. As with all papers reviewed by the journal, your manuscript was reviewed by members of the editorial board and by several independent reviewers. The reviewers appreciated the attention to an important topic. Based on the reviews, we are likely to accept this manuscript for publication, providing that you modify the manuscript according to the review recommendations.

It looks like we are very close to consensus on this paper. Reviewer two has a few last comments that need to be addressed.

Sincerely,

Alona Fyshe, Ph.D.

Associate Editor

PLOS Computational Biology

Wolfgang Einhäuser

Deputy Editor

PLOS Computational Biology

[LINK]

It looks like we are very close to consensus on this paper. Reviewer two has a few last comments that need to be addressed

Reviewer's Responses to Questions

**Comments to the Authors:**

Reviewer #1: The authors have addressed my concerns. I am happy to support publication.

Reviewer #2: Review of Xdream revision 1.

by Gary Cottrell

The reviewers have adequately addressed most of my comments, so this version is improved over the previous version, but there are still a few things that need to be addressed, and unfortunately, on the second reading, I have a couple of new questions for you.

I will also upload my marked-up version of the pdf for wording and typos I found.

I should say I only skimmed the supplementary material, but I did have one comment on the discussion: In the second paragraph, you refer to “the direct method”, which is not mentioned in the previous paragraph. You need to reintroduce this idea here, as a neuro person will not have any idea what you are talking about without going back to the main article.

In the paper, you linearly interpolate between the optimized images. In the response, you mention that you don’t do this in pixel space, but in the latent space. You should mention that here.

BTW, David Sheinberg showed some data at CNS in 2003 that may be relevant here. He told me it didn’t make it into a paper, but he found a cell in macaque that responded to both car images and butterfly images, with the strongest response to a particular car and a particular butterfly - very disjoint stimuli! It would be cool to stick that in here somewhere; I’ll upload the data with my review. I should mention that these images were very well known to the monkey.

There’s a typo on page 8/14, line 262, where you refer to Figure 3c, but I think you mean Figure 3d. On that same page, you again mention Table 1, without explanation. The explanation does appear in a caption under table (you should also mention the way you calculated the slope - I assume linear regression). Tables don’t have captions, so you will still need to move this into the main text, presumably before you mention the table.

Line 272 page 8:

With the “ivt” method, these initializations worked similarly in layers conv2 and conv4, not in layers fc6 (p=0.0014) and fc8 (p=2x10^-15).

This doesn’t say how they worked in layers fc6 anbd fc8 (again, these details are boring and irrelevant, since, as you say, PIs are not likely to have optimal images to start with. I still recommend leaving them to the supplementary material). But I’m not even sure what you are saying here. What are you referring to? Is it that opt is better than ivt in these layers? When I looked at the figure, I thought you were referring to how the best initialization gave bigger improvements here over medium and worst initializations for these layers. Please clarify this.

Likewise, the next sentence says that random initializations worked better, but you don’t say better than what, and it sure doesn’t look like they are better than the other results in Figure 3d, at least for medium and bad initializations.

Again, this paragraph doesn’t flow well. The message you want to get across is: In a realistic situation, where the PI doesn’t have access to optimal starting images, random initialization works about as well as anything else. Start by saying that starting with the optimal image does help in some cases, but given that investigators are not likely to have access to these, using random initialization is sufficient. Details in the supplementary material.

Figure 3d has some issues. In the caption, you give the order as best, middle, worst, but you’ve reversed the order from the previous manuscript - it’s now worst, middle, best. Also, the y-axis caption is incorrect - it should be “Relative Activation”, not “Target CaffeNet layer.”

Figure 4B has four different y-axis labels (!). This is not explained in the figure caption (or the text), so you may as well leave the extra three out, as they are confusing and a distraction from the main point.

Page 9: optimization methods: There are some significant differences between the algorithms. Can you comment on why you think that is?

Page 9: noise: There is a big difference of the effect of noise on the algorithm in the hidden layers versus the output. Any idea why? This seems odd, given that you’ve optimized your metaparameters on the output neurons, if I understood the methods correctly.

I trust that the editor can enforce these changes/clarifications; I don’t need to review this again.

**Have all data underlying the figures and results presented in the manuscript been provided?**

Reviewer #1: Yes

Reviewer #2: Yes

PLOS authors have the option to publish the peer review history of their article (what does this mean?). If published, this will include your full peer review and any attached files.

Reviewer #1: Yes: Michael Buice

Reviewer #2: Yes: Garrison W Cottrell
---

## [Editor Report · Decision Letter 2]

21 May 2020

Dear Mr. Xiao,

We are pleased to inform you that your manuscript 'XDream: finding preferred stimuli for visual neurons using generative networks and gradient-free optimization' has been provisionally accepted for publication in PLOS Computational Biology.

Best regards,

Alona Fyshe, Ph.D.

Associate Editor

PLOS Computational Biology

Wolfgang Einhäuser

Deputy Editor

PLOS Computational Biology

---

## [Editor Report · Acceptance letter]

9 Jun 2020

PCOMPBIOL-D-19-01642R2 

XDream: finding preferred stimuli for visual neurons using generative networks and gradient-free optimization

Dear Dr Xiao,

I am pleased to inform you that your manuscript has been formally accepted for publication in PLOS Computational Biology. Your manuscript is now with our production department and you will be notified of the publication date in due course.

With kind regards,

Laura Mallard
